# A De Novo Sequence Variant in Barrier-to-Autointegration Factor Is Associated with Dominant Motor Neuronopathy

**DOI:** 10.3390/cells12060847

**Published:** 2023-03-09

**Authors:** Agathe Marcelot, Felipe Rodriguez-Tirado, Philippe Cuniasse, Mei-ling Joiner, Simona Miron, Alexey A. Soshnev, Mimi Fang, Miles A. Pufall, Katherine D. Mathews, Steven A. Moore, Sophie Zinn-Justin, Pamela K. Geyer

**Affiliations:** 1Institute for Integrative Biology of the Cell (I2BC), CEA, CNRS, Université Paris-Saclay, 91198 Gif-sur-Yvette, France; agathecmarcelot@gmail.com (A.M.); philippe.cuniasse@cea.fr (P.C.); simona.miron@cea.fr (S.M.); 2Expression Génétique Microbienne, UMR 8261, CNRS, Institut de Biologie Physico-Chimique (IBPC), Université Paris Cité, 75005 Paris, France; 3Department of Biochemistry and Molecular Biology, Carver College of Medicine, University of Iowa, Iowa City, IA 52242, USA; felipe.rt@gmail.com (F.R.-T.); mei-ling-joiner@uiowa.edu (M.-l.J.); mimi-fang@uiowa.edu (M.F.); miles-pufall@uiowa.edu (M.A.P.); 4Department of Neuroscience, Developmental and Regenerative Biology, The University of Texas at San Antonio, San Antonio, TX 78249, USA; alexey.soshnev@utsa.edu; 5Department of Pediatrics and Neurology, Carver College of Medicine, University of Iowa, Iowa City, IA 52242, USA; katherine-mathews@uiowa.edu; 6Wellstone Muscular Dystrophy Specialized Research Center, Department of Pathology, Carver College of Medicine, University of Iowa, Iowa City, IA 52242, USA; steven-moore@uiowa.edu

**Keywords:** nuclear lamina, Barrier-to-autointegration factor, Néstor–Guillermo progeria syndrome, motor neuropathy, emerin, DNA binding, human variants

## Abstract

Barrier-to-autointegration factor (BAF) is an essential component of the nuclear lamina. Encoded by *BANF1*, this DNA binding protein contributes to the regulation of gene expression, cell cycle progression, and nuclear integrity. A rare recessive BAF variant, Ala12Thr, causes the premature aging syndrome, Néstor–Guillermo progeria syndrome (NGPS). Here, we report the first dominant pathogenic BAF variant, Gly16Arg, identified in a patient presenting with progressive neuromuscular weakness. Although disease variants carry nearby amino acid substitutions, cellular and biochemical properties are distinct. In contrast to NGPS, Gly16Arg patient fibroblasts show modest changes in nuclear lamina structure and increases in repressive marks associated with heterochromatin. Structural studies reveal that the Gly16Arg substitution introduces a salt bridge between BAF monomers, reducing the conformation ensemble available to BAF. We show that this structural change increases the double-stranded DNA binding affinity of BAF Gly16Arg. Together, our findings suggest that BAF Gly16Arg has an increased chromatin occupancy that leads to epigenetic changes and impacts nuclear functions. These observations provide a new example of how a missense mutation can change a protein conformational equilibrium to cause a dominant disease and extend our understanding of mechanisms by which BAF function impacts human health.

## 1. Introduction

The nuclear lamina lines the inner nuclear membrane. Composed of lamins- and lamin-associated proteins, this extensive network provides a structural platform for interactions with a wide variety of proteins [1,2,3]. Included in this network is Barrier-to-autointegration factor (BAF, sometimes referred to as BANF1 [4]), a 10 kDa, dimeric, sequence-independent, double-stranded DNA (dsDNA) binding protein that tethers chromosomes to the nuclear periphery. BAF has several direct nuclear lamina partners [5,6], including lamin A/C and proteins of the Lap2-emerin-MAN1 domain (LEM-D) family [7,8]. BAF enrichment at the nuclear periphery contributes to the regulation of gene expression, cell cycle progression, and maintenance of nuclear integrity [8,9,10,11].

BAF is essential for organism viability [5,11]. Complete loss of BAF causes a near absence of mitotic cells. Remaining mitotic mutant cells show defects in chromosome condensation, increased formation of anaphase bridges, defects in nuclear reformation, and altered nuclear structure [12,13,14,15]. In non-dividing cells, depletion of BAF induces aberrant nuclear lamina structure and partial chromatin clumping [13,15,16]. These observations emphasize that BAF is essential for nuclear organization and genome integrity throughout the cell cycle.

BAF DNA-binding activity is regulated by protein phosphorylation [17,18,19,20,21,22]. Unphosphorylated BAF interacts with DNA. As each monomer carries a DNA binding site, BAF dimers have the capacity to cross-link DNA [5,23,24,25]. Phosphorylation of BAF strongly impairs DNA binding [17,18,19,20,22,25,26]. Phosphorylation occurs at serine and threonine residues adjacent to the N-terminal helix α1 through the action of the mitotic Vaccinia-Related Kinases (VRKs [17,18,20]). Reversal of phosphorylation depends upon two phosphatases, including Protein Phosphatase (PP)2A, with its cofactor LEM4/ANKLE2 [27,28], and Protein Phosphatase (PP)4 [29,30,31]. Although crystal structures of the unphosphorylated and di-phosphorylated BAF are similar, di-phosphorylation induces a large conformational change in solution. The BAF N-terminal region that includes helix α1 is flexible in the unphosphorylated state but is significantly more rigid in the di-phosphorylated state due to salt bridges between the phosphorylated residues and the positively charged helix α6 [22,32]. We proposed that di-phosphorylated BAF adopts a more closed conformation, similar to that selected in the crystal state. Di-phosphorylation decreases BAF–DNA binding through two mechanisms. First, di-phosphorylation introduces four negative charges at the DNA binding site, causing electrostatic repulsion with the negatively charged sugar-phosphate backbone of DNA. Second, di-phosphorylation stabilizes a conformation that sterically clashes with DNA. However, it does not alter interactions with lamin A/C and emerin [22,32]. These observations suggest that changes in BAF structure can impact one of its functions without affecting others.

BAF influences higher-order chromatin structure through direct interactions with histones and chromatin modifying proteins [6,33]. Over-expression of BAF in cultured human embryonic kidney cells alters the landscape of histone modifications [34], significantly decreasing histone acetylation while increasing silencing marks of H3K27me and H3K9me. Such observations suggest that increased levels of BAF bound to DNA enhance recruitment of its chromatin modifying partners, such as G9a [6,34], the histone methyltransferase that is partially responsible for H3K9 methylation. Together, these findings emphasize the importance of BAF as an epigenetic regulator. 

The *BANF1* gene shows little variation among sequenced human genomes [35,36,37]. This limited variation is supported by its high loss-of-function intolerant score (gnomAD pLI 0.65, LOEUF 0.87; [38]). Of 23 variant *BANF1* alleles identified, nearly all occur in a heterozygous state, as single missense alleles [37]. Nevertheless, three individuals have been identified who carry a homozygous mutation of a *BANF1* allele that encodes the Ala12Thr variant [16,39]. These individuals have a recessive premature aging syndrome called Néstor–Guillermo progeria syndrome [NGPS], which affects tissues that depend upon adult stem cells. The mechanistic basis of NGPS is being clarified, with evidence indicating that the progeroid mutation disrupts a critical interaction between BAF and lamin A/C, without effect on its three-dimensional structure or DNA-binding activity [22,40]. 

Here, we report the identification and characterization of the first dominant pathogenic BAF variant, Gly16Arg. This de novo variant was identified in a patient displaying progressive neuromuscular weakness. Unlike findings in patients with NGPS, patient fibroblasts retain lamins and emerin at the nuclear periphery and show modest changes in nuclear shape. Biochemical studies reveal that the Gly16Arg variant has an altered structure characterized by significantly reduced dynamics of its N-terminal region due to the formation of an inter-monomer salt-bridge. Such changes increase its DNA binding affinity and elevate levels of repressive chromatin in patient fibroblasts. Based on these data, we predict that the Gly16Arg mutation alters the balance of BAF bound to DNA, leading to aberrant epigenetic changes that impact nuclear functions.

## 2. Materials and Methods

### 2.1. Cell Culture and Immunohistochemical Analysis

Patient fibroblasts (PL191444) and fibroblasts from an unaffected adult (SM54925) were derived from skin biopsies. Both fibroblast lines were provided by a cell culture repository that is a shared resource component of the Iowa Senator Paul D. Wellstone Muscular Dystrophy Specialized Research Center (MDSRC). These cultured cells are available for research as approved by the local institutional research ethics review board of the University of Iowa, IRB protocol #200510769, title “Muscle Biopsy/Cell Repository/Diagnostics Core”, original approval date 16 February 2006, latest continuing review approval date 11 July 2022. Primary cells were grown in DMEM with 20% FBS (GIBCO) supplemented with 1× glutamine, 1× non-essential amino acids, and 1× penicillin/streptomycin (GIBCO). Cells were grown to a confluence of less than 80% before splitting. At specific passages, 20,000 cells were seeded on coverslips. After 2 days of growth, cells were fixed with PFA 4% for 10 min at 4 °C. Cells were washed in PBS and permeabilized by incubation in 0.3% Triton X-100 for 10 min. For blocking, cells were incubated in 5% donkey serum in PBS for 1 h at 4 °C. Cells were incubated with primary antibodies overnight at 4 °C. Samples were washed and incubated for 2 h at room temperature with Alexa-Fluor conjugated secondary antibodies (1:500 dilution; Molecular Probes). Finally, coverslips were mounted using SlowFade Diamon Antifade Mountant with DAPI. Secondary-only controls were routinely run for all experiments. Side-by-side visualization with constant parameters for each antibody was performed using a Zeiss LSM710 confocal microscope and analyzed using ImageJ software v 1.53t. Primary antibodies included: (1) goat anti-Lamin B1 at 1:500 (B-10, Santa Cruz, Dallas, TX, USA); (2) mouse anti-emerin at 1:100 (Leica Biosystems, Deer Park, IL, USA clone 4G5); (3) mouse anti-Lamin A/C (Leica Biosystems, clone 636) at 1:50; and (4) rabbit anti-Histone H3K9me3 at 1:1000 (Abcam, Boston, MA, USA).

### 2.2. Image Quantification

Nuclear area and shape were measured in P10 and P15 cells. Nuclear area measurements were carried out using images of 100 randomly selected nuclei from two data sets using the measure object size function in CellProfiler software v. 4.2.1 [41]. To determine the overall percentage of lobulated or misshapen nuclei in cells at the different passage numbers, 100 nuclei per passage were randomly selected, and lobulation was scored as described in [42]. Briefly, lobulation was determined using Lamin B1 as a marker and defined as protrusions of Lamin B1 staining into the nuclear interior or an indentation between adjacent lobules. These determinations were made using a double-blind approach, followed by averaging of four data sets.

### 2.3. Western Blots

Western blots were performed as described previously [43]. Briefly, cell pellets were resuspended in Laemmli loading buffer, boiled for 3 minutes, and an equivalent of 10,000 cells per lane was resolved on 4–20% Tris-Glycine gel (Novex WedgeWell, Invitrogen XP04202). Proteins were transferred to a methanol-activated 0.45-micron PVDF membrane in Towbin buffer containing 20% methanol. After visualization with DirectBlue71 (212407, Sigma, Burlington, MA, USA), membranes were blocked for 2 hours with 5% nonfat milk in TBS-0.05% Tween20 (TBS-T) and incubated overnight with primary antibodies diluted at 1:1000 (α-H3K9me2, Abcam ab1220; α-H3K9me3, Abcam ab8898; α-H3K27me2, Cell Signaling 9728; α-H3K27me3, Cell Signaling 9733; α-GAPDH, Abcam ab8245), washed with TBS-T and incubated for 2 hours with HRP-conjugated secondary antibodies (AffiniPure alpaca α-mouse (615-035-214) and AffiniPure alpaca α-rabbit secondary antibodies (611-035-215), Jackson ImmunoResearch, West Grove, PA, USA) were diluted at 1:20,000 in 5% nonfat milk in TBS-T. After the final washes with TBS-T, HRP signals were visualized with SuperSignal West Pico PLUS chemiluminescent substrate (Thermo 34580) using a BioRad ChemiDoc system. Li-COR Image Studio Lite 5.2 software was used to quantify band intensities across three independent biological replicates assayed within linear range of HRP signal and normalized to the corresponding loading control (GAPDH).

### 2.4. Protein Constructs and Expression Vectors

BAF WT and BAF Gly16Arg were expressed as fusion proteins containing a His-tag (8 histidines), a TEV cleavage site. The human BAF sequence (either WT or mutated at Gly16) had all cysteines replaced with alanines in order to protect the protein from oxidation and thus, aggregation, as previously reported [44]. Genscript synthesized the BAF Gly16Arg gene after codon optimization for expression in *E. coli* in a pET M13 vector (kanamycin resistant). The lamin A/C IgFold domain corresponded to human lamin A/C residues 411 to 566, and was purified as previously reported [44], using a plasmid given by C. Östlund and H.J. Worman (pGEX vector; ampicillin resistant). The VRK1 catalytic domain corresponded to human VRK1 residues 3 to 364 and was purified as previously reported [22], using a plasmid given by John Chodera, Nicholas Levinson, and Markus Seeliger (Addgene plasmid # 79684; http://n2t.net/addgene:79684, accessed on 2 February 2023).

### 2.5. BAF WT and BAF Gly16Arg Protein Expression

Bacteria were grown in a rich medium (lysogeny broth, LB) for expressing proteins dedicated to ITC experiments, or either ^15^N or ^15^N/^13^C labeled M9 minimum media for expressing proteins dedicated to NMR experiments. Media were supplemented with 30 mg/mL kanamycin. A total of 20 mL of preculture inoculated 800 mL of culture (LB or M9). Expression was induced with 0.5 mM ITPG at an optical density OD_600_ = 0.8 and bacteria were then incubated at 20 °C overnight. Cells were harvested by centrifugation, flashed frozen in 30 mL of lysis buffer (50 mM Tris HCl pH 8, 300 mM NaCl, 5% glycerol, 0.1% Triton X-100, 1 mM PMSF), and stored at −20 °C for a maximum of 1 month.

### 2.6. Protein Purification

Both BAF WT and BAF Gly16Arg are insoluble after overexpression in *E. coli*; thus, purification was performed in urea and followed by a refolding step. After sonication in lysis buffer (50 mM Tris pH 8, 300 mM NaCl, 5% glycerol, 0.1% TritonX100), and centrifugation at 50,000× *g* for 15 min at 4 °C, the pellet was resuspended in urea purification buffer (50 mM Tris pH 8.0, 150 mM NaCl, 8 M urea) for 20 min. Then, the sample was re-centrifuged, and the soluble fraction incubated on Ni-NTA beads preequilibrated with urea purification buffer for 30 min at room temperature. Ni-NTA beads were washed with the purification buffer and the protein was eluted in 50 mL of the same buffer supplemented with 1 M imidazole. Proteins were then refolded by dialysis in BAF buffer (50 mM Tris pH 8, 150 mM NaCl). After concentration, the histidine tag was cleaved by the TEV protease (from a batch of TEV purified in the lab) overnight at 4 °C. Proteins were separated from the TEV protease (containing a histidine tag), and its tag by Ni-NTA affinity chromatography. Finally, gel filtration was performed using a Superdex 200 pg HiLoad 16/600 column (GE healthcare; Appendix A). The final yield was typically about 0.6 mg (LB) or 0.1 mg (M9) of purified protein per liter of bacterial culture for each BAF constructs.

For purification of the lamin A/C Igfold domain, after bacteria were sonicated at 10 °C, the supernatant was incubated 20 min at room temperature with benzonase (250 u) and centrifuged at 50,000× *g* for 15 min at 4 °C. The soluble extract was supplemented with 5 mM DTT and loaded onto glutathione beads. After 1 h of incubation at 4 °C, glutathione beads were washed with 1 M NaCl buffer and then with the purification buffer (50 mM Tris pH 7.5, 150 mM NaCl, 5 mM DTT). The GST tag was cleaved with thrombin (commercial thrombin from Sigma Aldrich), at 200 units per mL for 2 h at room temperature. Protein was recovered in the flow-through and separated from thrombin and last contaminants using gel filtration (Superdex 200 pg HiLoad 16/600 column from GE healthcare; Appendix A). The final yield was typically 20 mg (LB) or 6 mg (M9) of purified protein per liter of bacterial culture.

VRK1 is soluble after bacterial expression. After sonication, the soluble extract was incubated with benzonase (250 u) for 20 min at 20 °C. The lysate was centrifuged at 50,000× *g* for 15 min at 4 °C and loaded onto a 5 mL Ni-NTA column (FF crude, GE-Healthcare). The column was washed with 50 mM Tris pH 8.0, 1 M NaCl, preequilibrated with 50 mM Tris pH 8.0, 150 mM NaCl), and the protein was eluted using an imidazole gradient (0 to 500 mM). After concentrating the eluate to 5 mL, the histidine tag was cleaved by TEV protease (from a batch of TEV purified in the lab) for 1.5 h at 20 °C. Proteins were separated from the His-tagged TEV protease by affinity chromatography, using Ni-NTA beads. A final gel filtration (Superdex-200 HiLoad 16/600 column) step was performed in 50 mM Hepes pH 7.5, 150 mM NaCl to remove the last contaminants. The final yield was typically 28 mg (LB) of purified protein per liter of bacterial culture.

### 2.7. Liquid-State NMR Spectroscopy

Most NMR experiments were performed on a 700 MHz Bruker AVANCE NEO spectrometer equipped with a triple resonance cryogenic probe. A set of NMR assignment experiments were performed on a 600 MHz Bruker Advance II spectrometer equipped with a triple resonance cryogenic probe. The data were processed using Topspin v. 4.0.2 to v. 4.0.8 (Bruker) and analyzed using Topspin 4.1.3 (Bruker) and CCPNMR 2.4 [45]. Sodium trimethylsilylpropanesulfonate (DSS) was used as a chemical shift reference and 3 to 5% of D_2_O were added to the NMR samples to ensure the lock. BAF WT ^1^H-^15^N chemical shifts were assigned in our previous study [22]. In order to assign the BAF Gly16Arg ^1^H-^15^N chemical shifts, 3D heteronuclear NMR experiments (3D HNCO, HNCACO, HNCA, and CBCACONH) were recorded in the same conditions as for BAF WT, i.e., using a sample containing about 350 µM of ^15^N/^13^C labeled protein in a 40 mM sodium phosphate buffer at pH 7.2, with 150 mM NaCl and 1× antiproteases (Roche), using a 3 mm diameter Shigemi tube, at 293 K. Both BAF WT and BAF Gly16Arg phosphorylation kinetics were monitored at 303 K on a 700 MHz spectrometer, using 3 mm diameter NMR tubes containing 150 µM of BAF protein and 150 nM of VRK1 kinase (molar ratio vs. BAF of 0.1%) in the NMR phosphorylation buffer (40 mM HEPES pH 7.2, 150 mM NaCl, 5 mM ATP, 5 mM MgSO4, 1 mM TCEP, 1× antiproteases (Roche)). Two-dimensional ^1^H-^15^N HSQC spectra were acquired every 20 min and one-dimensional ^1^H spectra were recorded in between each HSQC to report for potential pH drifts upon phosphorylation reaction. In order to characterize BAF WT and BAF Gly16Arg dynamics at the residue level, ^1^H→^15^N nOe experiments were recorded at 700 MHz, 293 K, in a 4 mm diameter Shigemi tube containing about 250 µM of protein in 40 mM sodium phosphate pH 7.2, 150 mM NaCl, 1× antiproteases (Roche). Interleaved nOe and reference experiments were performed using a relaxation delay of 6 s in triplicate for non-phosphorylated and di-phosphorylated BAF WT and in duplicate for BAF Gly16Arg.

### 2.8. ITC Binding Assays

Interactions between BAF WT or BAF Gly16Arg, and the lamin A/C IgFold domain (Appendix A) or a 22 nt-dsDNA (Appendix A) were analyzed using a VP-ITC calorimetry system (MicroCal-Malvern Panalytical, Malvern, UK). Calorimetric titrations were performed with 10–20 µM of monomeric BAF (WT or Gly16Arg) in the calorimetric cell and 100 µM of lamin A/C Igfold domain or 31–33 µM of dsDNA in the injecting syringe. All samples were in 50 mM Hepes pH 7.4 and 150 mM NaCl. All measurements were performed at 288 K because of the high signal-to-noise ratio observed at this temperature. For each titration, a sequence of 29 times 10 µL injections was programmed, with a reference power of 10 µcal/s and spacing between injections of 180 s. Experiments were performed in duplicate. Data were analyzed using Origin (OriginLab, Northampton, MA, USA) and the stoichiometry was set to 0.5 in the case of the micromolar interaction between BAF and lamin A/C IgFold domain, assuming each BAF dimer binds to one lamin A/C IgFold domain.

### 2.9. Molecular Dynamics (MD) Simulation

Three 1 μs MD simulations of the dimeric BAF Gly16Arg mutant were calculated, starting with the same structure but with randomized initial velocities. The initial structure was built from the BAF crystal structure (PDB code 1CI4, hereafter designated as the reference structure [46], after mutation of residues seleno-Met15 to methionine and Gly16 to arginine using the PyMol builder (Schrödinger L, Delano W. PyMOL. Available at: http://www.pymol.org/pymol (accessed on 2 February 2023). Because the BAF Gly16Arg was simulated as a homodimer without symmetry constraints, the total sampling time is equivalent to 6 times the length of a simulation with 6 different starting structures. We checked the completeness of sampling by calculating the configurational entropy as a function of the simulation time and verifying that it converges. Therefore, we removed overall rotations and translations that occur during the simulation by superimposing the extracted structures (frames from 20 ns to 1 μs) onto the BAF reference structure through minimization of the RMS deviation on the Cα positions in helix α3, α4, α5 and α6. Then, we calculated the configurational entropy of each monomer in the quasi-harmonic approximation [47]. 

All MD calculations were carried out using NAMD [48] in the CHARMM v36 force field [49]. The hydrogens lacking in BAF crystal structure were first built using the HBUILD function of CHARMM v42b2. The resulting model was immersed in a cubic water box using the TCL solvate plugin of VMD [50]. The size of the box was defined such as the minimum size of the water layer surrounding the solute was 12 Å. Then, the system was neutralized with chloride ions using the autoionize plugin of VMD. Periodic boundary conditions were calculated using Particle Mesh Ewald electrostatics (grid spacing of 1 Å). A cutoff of 12 Å was set for electrostatics and Lennard Jones calculations. These interactions were smoothened from 10 Å to 12 Å with a switching function. The dielectric constant was set to 1.0. Langevin dynamics were used, and the temperature was set to 310 K. The initial structure was submitted to an initial 30,000 steps energy minimization during which the harmonic potential applied to the position of the Cα and Cβ atoms was gradually decreased to 1 kcal.mol^−1^·Å^−2^. Then, a 1 ns equilibration step was run in constant-temperature and constant-pressure conditions. During this equilibration step, the coordinates of the Cα and Cβ atoms were restrained to their initial position using a harmonic potential with a force constant of 1 kcal.mol^−1^·Å^−2^. Then, the unrestrained production step was started for 1 μs in the same constant-temperature and constant-pressure conditions. Analysis of the trajectories was achieved using in-house scripts written in the macro language of CHARMM. The three MD trajectories were calculated with the NAMD 2.14 CUDA version on the TGCC HPC GPU resources made by GENCI (allocation 2021-A11; Grant No. A0110712964).

## 3. Results

### 3.1. Identification of a De Novo BANF1 Mutation in a Patient with Motor Neuronopathy

An 8-year-old girl was evaluated for progressive muscular weakness (Figure 1A). She is one of three children, and no family history of neurologic disease was reported (Figure 1B). She was born at term, was healthy at birth, and met early developmental milestones, including walking at 11 months. At age 3, an abnormal, clumsy gait was recognized. Her gait worsened with age, leading to neurologic evaluation at age 5. At that time, spine MRI and nerve conduction velocities were normal. Her EMG showed a chronic neurogenic process with diffuse fibrillations and reduced motor responses in distal legs. Nerve conduction velocities were normal. Over time, she experienced slowly progressive weakness and chronically poor weight gain.

She was referred to the University of Iowa Hospitals and Clinics (UIHC) at age 8 years. At UIHC, her examination showed low weight (<3rd percentile, height at 20th percentile), normal cognition, globally decreased muscle bulk, mild intention tremor, diffuse muscle weakness that affected legs more than arms, a complete foot drop and slightly decreased cold sensation of the toes (Figure 1A). She was slow in rising from lying to standing, taking 8–10 s using a full Gowers’ maneuver. Deep tendon reflexes were normal in the arms, easily elicited at the knees, but absent at the ankles. Tongue fasciculations were absent. The working diagnosis was a motor neuronopathy (anterior horn cell vs. motor axon). At age 9 years, forced vital capacity was 60% of predicted. At age 10 years, scoliosis developed. At age 12 years, she required a wheelchair full time. A gastrostomy tube was placed due to aspiration and poor weight gain, and nocturnal non-invasive ventilation was initiated. Serial exams were notable for progressive symmetric loss of deep tendon reflexes, worsening weakness affecting both proximal and distal muscles, and variably decreased cold sensation in the feet without other sensory abnormalities. At age 17 years, she required non-invasive daytime mouthpiece ventilation due to progressive weakness of muscles of respiration.

To clarify the cause of her motor neuronopathy, several commercial diagnostic genetic tests were performed. Axonal forms of Charcot Marie Tooth (CMT) disease are a group of diseases presenting with progressive weakness wasting [51]. An axonal CMT panel (GeneDx21 genes) did not identify any pathogenic variants. A comprehensive neuromuscular panel (InVitae, 104 genes) only identified a variant of unknown significance in *AGRN,* a gene critical for the development of the neuromuscular junction [52]. Next, trio (patient and parents) exome sequencing (GeneDx) was completed. These studies identified a de novo c.46G>A mutation in *BANF1*, a finding confirmed by direct PCR analysis of genomic DNA obtained from patient fibroblasts (Figure 1C). This mutation was absent in both parents and her two siblings (both unaffected), consistent with its association with disease. Notably, this transition mutation encodes a novel variant of BAF, in which an arginine residue replaces glycine 16 (Gly16Arg). The mutated amino acid residues in an evolutionarily conserved patch within BAF (Figure 1D), positioned near the dimer interface and the interface used in binding the lamin A/C IgFold domain (Figure 1E [44]). As expected for a causal variant, this missense mutation is absent from the ExAC and gnomAD databases [36]. Based on the changed biochemical nature of the amino acid substitution and its location, we predicted that this de novo missense mutation is pathogenic. 

### 3.2. The Gly16Arg Variant Confers Modest Changes to Nuclear Lamina Structure

To understand the cellular effects of the BAF Gly16Arg variant, a skin biopsy was obtained from the patient and used to establish a culture of primary dermal fibroblasts. Immunohistochemical analysis of early passage (P10) patient cells and control human skin fibroblasts was completed to assess effects of the BAF variant on two partner proteins, lamin B1 and emerin. These analyses showed normal nuclear lamina localization of both proteins (Figure 2A). These findings contrast with those of NGPS patient fibroblasts that express only BAF Ala12Thr, wherein nuclear accumulation of emerin is largely lost and emerin is found predominantly in the cytosol [16]. Differences in nuclear structure are consistent with the disparate phenotypic features of motor neuron disease versus progeria.

The Gly16Arg missense mutation lies near the BAF and lamin A/C interaction surface [44]. We tested whether this substitution alters BAF association with lamin A/C in vitro. To this end, we expressed and purified BAF wild type (WT) and its Gly16Arg variant. Size-exclusion chromatography of the two proteins showed that the elution profile of BAF Gly16Arg is nearly identical to that of BAF WT (Appendix A), indicating that the missense variant permits dimer formation. Nuclear Magnetic Resonance (NMR) analysis of the purified proteins showed that the 2D ^1^H-^15^N HSQC spectra are largely superimposable, indicating that they share the same fold (Appendix A). Next, we measured the affinity between BAF (either WT or Gly16Arg) and the lamin A/C IgFold domain by Isothermal Titration Calorimetry (ITC). Dissociation constants of BAF WT and BAF Gly16Arg binding to the lamin A/C IgFold domain measured 2.7 ± 0.3 μM and 9.0 ± 0.8 μM, respectively (Figure 2B; Appendix A), corresponding to a threefold decrease in affinity of Gly16Arg, a value that contrasts with the nearly 10-fold decrease in affinity shown by the recessive NGPS Ala12Thr variant [40]. To understand whether such a threefold reduction affects the cellular distribution of lamin A/C, we used immunohistochemical analysis of early (P10) and later (P15) passage cells. We found that the distribution of lamin A/C in patient fibroblasts was unaffected (Figure 2C). Nonetheless, these analyses revealed some changes in nuclear structure, including increased nuclear lobulation and decreased nuclear area (Figure 2D,E). However, the severity of these nuclear changes is minimal relative to those found in NGPS cells [16] and patient cells of a second progeroid syndrome Hutchinson Guilford Progeria (HGPS) [42]. Taken together, we conclude that a modest reduction in the lamin A/C interaction is not sufficient to explain the dominant disease caused by BAF Gly16Arg.

### 3.3. The BAF Gly16Arg Structure Is Intermediate to Un- and Di-Phosphorylated BAF WT

BAF localization and function are strongly regulated by VRK1. For this reason, we analyzed whether the Gly16Arg mutation impairs BAF phosphorylation in vitro. We previously showed that VRK1 phosphorylates BAF WT first on Ser4 and then on Thr3 [22]. Using NMR spectroscopy, we similarly monitored the phosphorylation of BAF Gly16Arg at the residue level. After the addition of VRK1, we found signals appearing that correspond to pSer4, and then pThr3 in the BAF Gly16Arg spectrum (Appendix A). Moreover, the phosphorylation kinetics were similar to that of BAF WT. We conclude that the Gly16Arg mutation does not impair BAF phosphorylation. 

A detailed comparison of the ^1^H-^15^N HSQC NMR spectra of unphosphorylated and di-phosphorylated BAF WT and BAF Gly16Arg was undertaken. We draw two main conclusions from our NMR analysis. First, ^1^H-^15^N HSQC peaks of unphosphorylated BAF Gly16Arg are located in between peaks of unphosphorylated and di-phosphorylated BAF WT (Figure 3A; Appendix A). ^1^H-^15^N HSQC peaks from these three BAF species are generally aligned, as shown for glycine residues, Asp40, Ala42, Lys53, Lys54, and Ala77 in Figure 3A. The main exceptions are peaks corresponding to residues close to the phosphorylated residues, whose NMR signals are perturbed by the presence of the phosphate groups. Second, ^1^H-^15^N HSQC peaks of di-phosphorylated BAF Gly16Arg are close to those of di-phosphorylated BAF WT, so that the ^1^H-^15^N HSQC NMR signals of the four BAF species are mostly aligned (Figure 3A). Such observations reveal that the four species are in exchange between two ensembles of structures, defined as open and closed states [22,32]. The open conformation, adopted by BAF WT, is characterized by a highly flexible N-terminal region including a transiently folded helix α1, whereas the closed conformation, induced by phosphorylation, shows a more folded helix α1 positioned closer to helix α6. Salt-bridges between the phosphorylated residues and positively charged residues from helix α6 stabilize the closed conformation [22]. Our NMR analysis of BAF Gly16Arg phosphorylation by VRK1 indicates that unphosphorylated BAF Gly16Arg adopts an ensemble of conformations that is more restricted than BAF WT, but more dispersed than di-phosphorylated BAF, whereas di-phosphorylated BAF Gly16Arg adopts a structure similar to that of di-phosphorylated BAF WT. To further investigate BAF Gly16Arg dynamics, we measured ^1^H → ^15^N heteronuclear nOes and compared them to the previous data obtained on unphosphorylated and di- phosphorylated BAF WT [22]. Our previous studies reported that ^1^H → ^15^N heteronuclear nOes values are low in the N- terminal region of BAF WT as compared to di-phosphorylated BAF WT, indicating that the BAF WT N-terminus is significantly more flexible on a picosecond to nanosecond timescale. In the case of BAF Gly16Arg, ^1^H→^15^N heteronuclear nOes values measured for three residues of helix α1 (Lys6, Arg8, Asp9) are intermediate between values measured for BAF WT, either non phosphorylated or di-phosphorylated, confirming that the flexibility of this helix is significantly impacted by the mutation (Figure 3B). Together, our data support that unphosphorylated BAF Gly16Arg adopts an ensemble of conformations in solution in which helix α1 is less flexible as compared to BAF WT, but more flexible as compared to di-phosphorylated BAF WT or Gly16Arg (Figure 3C).

### 3.4. Detection of an Inter-Monomeric Salt-Bridge Involving Arg16 Using In Silico Analyses

To further describe the structural differences between the N-terminal regions of BAF WT and BAF Gly16Arg, we performed a MD simulation, following the protocol we previously used for studying unphosphorylated and di-phosphorylated BAF WT (Figure 4A–C [32]). Thus, we calculated the conformational ensembles accessible to BAF Gly16Arg at an atomic scale, which we described by calculating the Root-Mean-Square (RMS) deviation of the position of the Cα along the MD trajectories with respect to the BAF crystal structure. The distribution of the RMS deviation for residues 1 to 12 reflects the disorder of the amino-terminal region of BAF (Figure 4D–F). Focusing of the N-terminal region revealed no obvious difference due to mutation Gly16Arg. Closer examination of residues 1 to 4 showed that they are highly flexible in both BAF WT and Gly16Arg, which is consistent with the two N-termini being accessible to VRK1 for phosphorylation (Figure 4G–I). However, the conformational ensemble accessible to helix α1 (residues 5 to 12) is significantly reduced in the Gly16Arg variant (Figure 4J–L). Indeed, the RMS deviation on the position of the Cα of residues 5 to 12 with respect to BAF crystal structure displays a unimodal distribution centered on 3.0 Å for BAF Gly16Arg, whereas it shows a bimodal distribution with one peak centered at 3.0 Å and one at 5.0 Å for BAF WT. Further, the distribution of the RMS deviation of the Cα for residues 5 to 12 in the Gly16Arg mutant resembles the one calculated for di-phosphorylated BAF, although additional conformers with RMS deviations below 2.0 Å were observed in phosphorylated BAF WT. These MD simulation results support our NMR analysis. Next, we examined the inter-residue interactions along the MD trajectories. We identified a salt-bridge interaction between the guanidinium group of Arg16 of one monomer and the C-terminal carboxyl group of Leu89 of the other monomer (Figure 4M). This interaction was not present in the initial structure and was rapidly formed for both monomers and was stable along the three 1 µs MD trajectories calculated for BAF Gly16Arg (Figure 4N). Altogether, our NMR and molecular dynamics analyses reveal that the conformational ensemble of residues 5 to 12 transiently forming helix α1 is significantly narrower when Gly16 is mutated into arginine. We suggest that the stabilization of helix α1 is due to a salt-bridge formed between the guanidinium group of Arg16 of one monomer and the C-terminal carboxyl group of Leu89 of the other monomer (Figure 4M), which constrains the conformation of BAF in solution.

### 3.5. BAF Gly16Arg Shows Increased DNA Binding Affinity

VRK1 phosphorylation of the N-terminal region of BAF (Thr3, Ser4) strongly decreases the affinity of BAF for dsDNA (Figure 5A; [17,18,19,20,21,22]). This decrease results from both electrostatic repulsion between phosphorylated residues and DNA, and a phosphorylation-induced shift of the conformational ensemble of the N-terminal region of BAF that causes a steric clash with dsDNA [32]. Because mutation Gly16Arg modifies the conformational ensemble of the N-terminal region of BAF, we hypothesized that the DNA binding affinity of this variant might change. To this end, we measured the affinity of BAF Gly16Arg for a 22-nt dsDNA using ITC (Appendix A). We found that the DNA binding free energy of the mutant consists of an unfavorable decrease in the enthalpic contribution, changing from −49.5 kcal/mol in BAF WT to −18.1 kcal/mol for BAF Gly16Arg (Appendix A). However, this negative contribution is offset by a large and favorable decrease in the entropic penalty from 40.4 kcal/mol to 7.81 kcal/mol. Addition of these two components results in a free energy of −9.1 kcal/mol versus −10.3 kcal/mol for BAF WT and BAF Gly16Arg, respectively. Thus, there is a sevenfold increased affinity to dsDNA of BAF Gly16Arg as compared to BAF WT, with Kd values of 16.3 ± 5.7 nM and 117± 58 nM, respectively (Figure 5B, Appendix A). An increased dsDNA binding affinity of BAF Gly16Arg is consistent with its dominant gain-of-function disease phenotype.

Each BAF monomer binds DNA (Figure 5A), leading to DNA cross-linking by the dimer [14,46]. Building from the observations of increased DNA binding affinity in the BAF Gly16Arg, we postulated that expression of the Gly16Arg variant might increase chromatin compaction in vivo, evidenced by expansion of heterochromatin. To this end, we stained patient and control fibroblasts with antibodies against tri-methylated lysine 9 on histone H3 (H3K9me3), a histone post-translational modification uniformly associated with heterochromatin. We observed significant increases in H3K9me3 in both early and later stage passages (Figure 5C). To extend these observations, we performed Western blot analyses on proteins extracted from early and late-stage cell passages. Specifically, we investigated the levels of H3K9me2, H3K9me3, H3K27me2, and H3K27me3, with these latter two types of histone post-translational modifications associated with facultative heterochromatin. These analyses showed increased levels of H3K9me3 and H3K27me2 in late passage (P15) BAF Gly16Arg fibroblasts (Figure 5D, Appendix A). Our findings suggest that the increased DNA binding affinity of BAF Gly16Arg leads to an altered chromatin occupancy in vivo, resulting in an aberrant epigenetic landscape. Notably, these findings differ from those observed in NGPS and HGPS cells that carry a lamin A/C mutation [40,42,53], emphasizing differences in the effects of BAF Gly16Arg and mutations that lead to progeroid diseases.

## 4. Discussion

BAF is a conserved metazoan chromatin protein that has critical roles in nuclear organization and genome integrity. Nearly all variant encoding alleles are found in a heterozygous state [37], arguing that complete loss of BAF is lethal, consistent with findings in model organisms that BAF is essential [12,13]. Nonetheless, a homozygous recessive mutation of *BANF1* was identified in three individuals with NGPS, causing a rare type of progeria [16,39]. Notably, the NGPS BAF variant retains its DNA binding capacity, but has reduced lamin A/C association [22,40]. These observations illustrate that *BANF1* separation of function mutations exist. 

Here, we report the identification of an individual with progressive motor neuron disease that carries a dominant mutation in *BANF1*. The mutation is novel, leading to a change in the Gly16 residue to Arg (Figure 1C,D). Although the mutated amino acid is located four residues away from the NGPS Ala12Thr mutation, clinical disease features and cellular phenotypes differ. For example, NE localization of emerin and lamins in Gly16Arg patient fibroblasts is unchanged, and nuclear structure is largely normal (Figure 2A,C–E), whereas Ala12Thr patient cells show cytoplasmic accumulation of emerin and nuclear blebs [16]. Consistent with an unchanged lamin distribution, our biochemical studies demonstrate that although interactions between BAF Gly16Arg and the lamin A/C IgFold domain are significantly compromised, these effects are weak (Figure 2B), contrasting with the strong reduction for the NGPS BAF variant measured using the same methods [44]. Further, the modest reduction in lamin A/C affinity does not explain a dominant disease phenotype. Taken together, we conclude that the Gly16Arg mutation affects a function of BAF that differs from that affected by the nearby Ala12Thr mutation.

Our NMR and MD simulation indicate that BAF Gly16Arg has an altered protein structure, largely manifest in the N-terminal region of the protein (Figure 3 and Figure 4). In the crystal state, BAF variants that include BAF with cysteines mutated into alanine and phosphorylated BAF adopt the same 3D structure [22]. In contrast, in solution, BAF exhibits a highly flexible N-terminal region, including a transiently folded helix α1 from Gln5 to Ala12. NMR analyses show that the conformation ensemble described by residues 5 to 12 of BAF Gly16Arg is significantly reduced relative to that of BAF WT, findings that are supported by MD simulations that consistently found decreased dynamics. Further, MD simulations suggest that this results from the formation of a stable salt-bridge between the guanidinium group of the mutant Arg16 of one monomer with the C-terminal carboxylate of Leu89 of the other monomer (Figure 4M,N). The presence of a cross-monomer interaction restricts the conformational ensemble of BAF Gly16Arg in solution, and more specifically decreases the flexibility of residues 5 to 12. The resulting conformational ensemble described by residues 5 to 12 in BAF Gly16Arg shifts towards conformations found in di-phosphorylated BAF WT and Gly16Arg.

Even though mutant BAF Gly16Arg adopts a structure reminiscent of a non-DNA binding form of BAF, this variant has an increased ds-DNA binding affinity (Figure 5B). Our data suggest that the increased conformational order in BAF Gly16Arg reduces the entropic penalty of binding to DNA, leading to a sevenfold increase in affinity (Figure 5B, Appendix A). We envision that this change has in vivo consequences, a prediction supported by immunohistochemical and Western analyses. We show that patient fibroblasts have increased levels of histone H3K9me3 and H3K27me2 relative to that of control cells, two markers associated with condensed, repressive heterochromatin (Figure 5C,D). Similar increases in levels of these heterochromatic markers were found in cells over-expressing BAF [34], indicating that increasing the DNA occupancy of BAF is capable of altering the levels of histone modifications. We suggest that increased DNA occupancy might promote heterochromatin formation through direct interactions between BAF and histones [33]. Alternatively, enhanced chromatin association of BAF Gly16Arg might result in increased recruitment of BAF interaction partners, such as the histone H3 methyltransferase G9a that is partially responsible for the deposition of the H3K9me3 marker, or histone deacetylases [6,34]. Based on our findings of increased marks associated with heterochromatin formation, we predict that dominant effects of BAF Gly16Arg change the epigenetic and transcriptional landscape of the cell.

Increased BAF–DNA binding might account for other neurological diseases. Mutations in *VRK1* [54], the kinase that phosphorylates BAF during mitosis, are associated with motor neuron disease with or without associated features such as pontocerebellar hypoplasia or intellectual disability [55]. Reported cases of VRK1-related motor neuron disease without associated features have a phenotype similar to the patient reported here. Additionally, variation of the gene encoding a second BAF kinase, *VRK2*, is associated with several neurological disorders [56]. These observations, coupled with our findings on the effects of the BAF Gly16Arg variant, suggest that regulation of BAF chromatin association might be critical for neuronal health. A link between altered BAF function and neurodegenerative disease extends the consequences of BAF dysfunction as it pertains to human disease.

## Figures and Tables

**Figure 1 cells-12-00847-f001:**
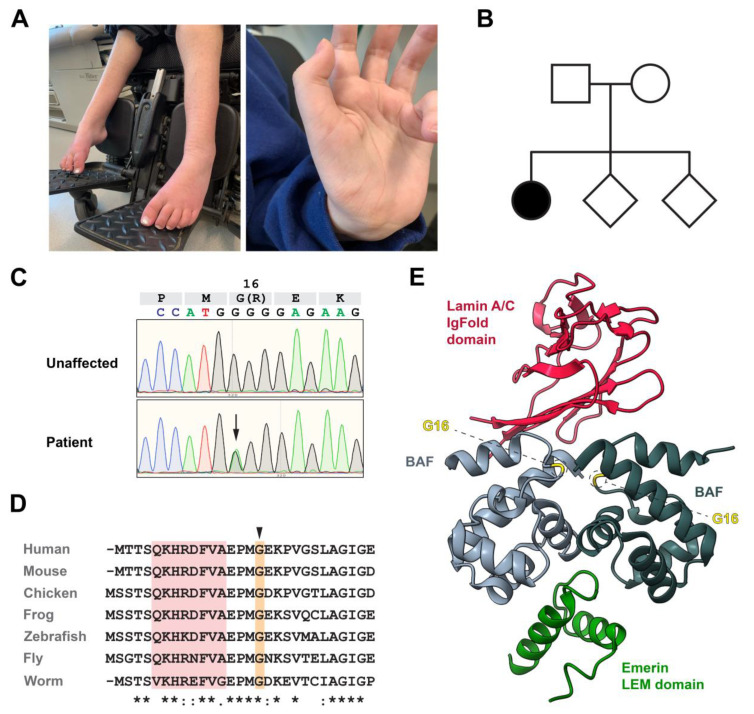
Identification of a de novo BANF1 mutation. (**A**) Photographs of hands and lower extremities of patient at 17 years, showing diffuse muscle weakness affecting legs. Contracture and drop of the feet are evident. Hand contractures are also observed. (**B**) Pedigree of the patient family, wherein the affected individual is shown as the black circle and unaffected individuals are shown as open shapes. (**C**) Direct sequence analysis of genomic DNA extracted from control and patient fibroblasts, confirming the heterozygous c.46G to A substitution (black arrow). (**D**) Sequence alignment of the first 28 residues of human BAF to the orthologs of the indicated species. Identical (*) and chemically similar (:) amino acids are noted below. The position of the mutated Gly is shown by the arrowhead above the alignment. The position of helix α1 is shown by the red background. (**E**) Crystal structure of the complex between the BAF dimer (monomers shown in light and dark grey), the lamin A/C IgFold domain (red) and the emerin LEM domain (green). The position of the mutated Gly in each BAF monomer is indicated. The PDB reference of this 3D structure is 6GHD.

**Figure 2 cells-12-00847-f002:**
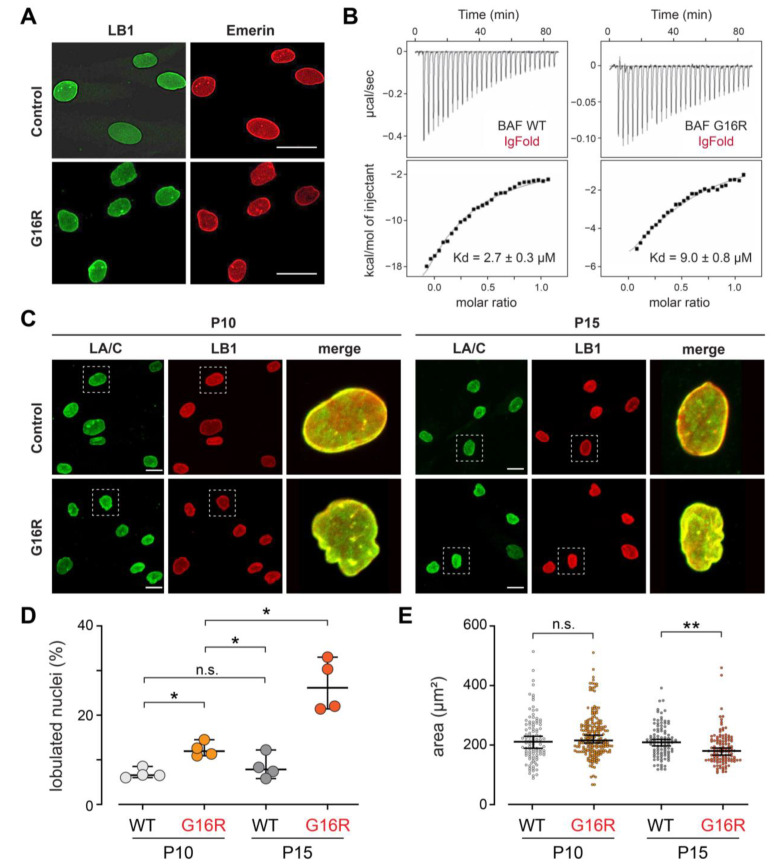
Reduced LA/C association with BAF Gly16Arg confers modest nuclear shape changes. (**A**) Confocal microscope images of control and patient passage 10 fibroblasts stained for lamin B1 (LB1; green) and emerin (red). Scale bars are 20 µm. (**B**) ITC titration analyses providing the affinity of BAF WT and BAF Gly16Arg for the lamin A/C IgFold domain. The experiment was repeated twice, and the dissociation constants represent the mean values calculated from these experiments (see Appendix A). (**C**) Confocal microscope images of control and patient passage 10 and 15 fibroblasts stained for lamin A/C (LA/C; green) and lamin B1 (LB1; red). An enlarged image of the boxed nucleus is shown on right hand side of panel. Scale bars are 20 µm. (**D**,**E**) Graphs of changes in (**D**) shape and (**E**) nuclear area at P10 and P15. For nuclear shape, nuclei were scored as lobulated if they contained >1 lobulation [42]. Determinations were made using a double-blind approach, followed by averaging four data sets. For each coverslip, 100 nuclei were scored. Averages were compared using the Mann–Whitney test for non-parametric data. * *p* < 0.05. For nuclear area, measurements were made on 100 nuclei plated on two independent coverslips using CellProfiler software [41]. Means were compared using the unpaired *t*-test. ns = not significant, ** *p* < 0.01.

**Figure 3 cells-12-00847-f003:**
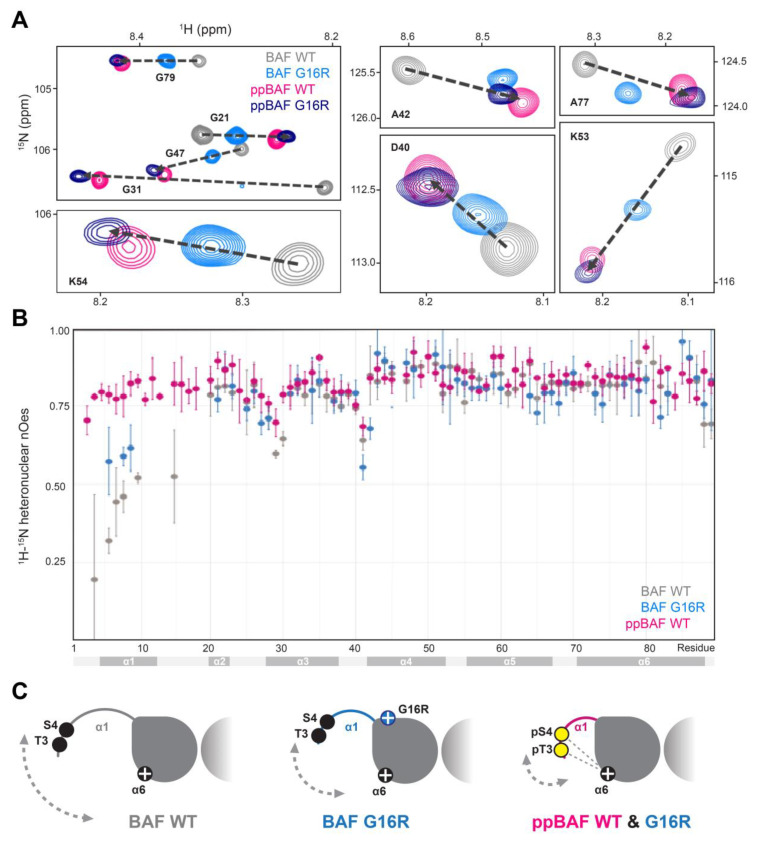
The Gly16Arg mutation modifies solution conformations of BAF by reducing flexibility of the amino terminus. (**A**) Zoomed-in views of the superimposition of the 2D NMR ^1^H-^15^N HSQC spectra of BAF WT (in grey), BAF Gly16Arg (in cyan), di-phosphorylated BAF WT (in hot pink), and di-phosphorylated BAF Gly16Arg (in dark blue). Full views are displayed in Appendix A. The NMR signals of the four species are aligned, showing that they are all exchanging between the same two ensembles of conformations, defined as open and closed states in [22,32]. (**B**) Plot of the NMR ^1^H→^15^N nOe values as a function of the residue number (BAF WT: grey; BAF Gly16Arg: cyan; di-phosphorylated BAF WT: hot pink). These values reflect the dynamics on a fast (picosecond to nanosecond) timescale of the amide bonds of BAF residues. Low and high values indicate flexibility and rigidity, respectively. (**C**) Scheme illustrating that BAF Gly16Arg exhibits an N-terminal region fluctuating between open and closed conformations, which is positioned more often in a closed conformation when compared to BAF WT, but more often in an opened conformation when compared to di-phosphorylated BAF WT. The more rigid helix α1 region of BAF Gly16Arg compared to WT is indicated by a bold blue line, whereas the salt-bridges restraining the position of the N-terminus relatively to helix α6 in di-phosphorylated BAF WT and Gly16Arg are marked with dotted lines. The Gly16Arg mutation modifies solution conformations of BAF by reducing flexibility of the amino terminus.

**Figure 4 cells-12-00847-f004:**
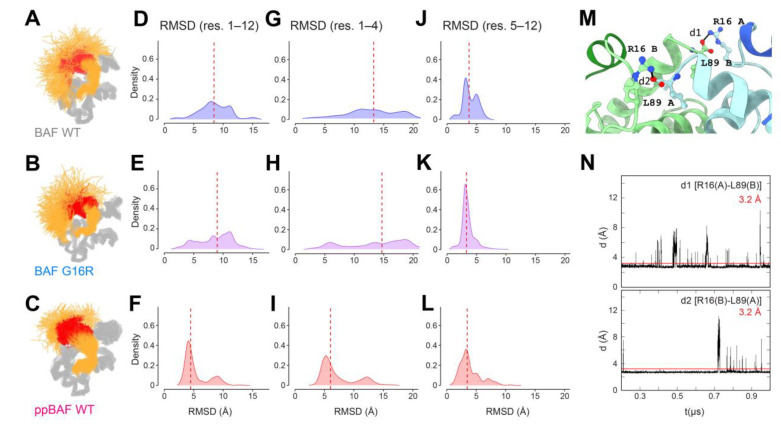
The structure of helix α1 in BAF Gly16Arg is less variable than in BAF WT. (**A**–**C**) Superimposition of a set of 6000 monomeric structures extracted from the MD trajectories of BAF WT (BAF; [32]), BAF Gly16Arg and di-phosphorylated BAF WT (ppBAF; [32]). Residues 20–89 are colored in grey and residues 1 to 19 are colored in orange except for residues 5 to 12 (transiently forming helix α1) that are colored in red. The structures were superimposed on the core region of the reference BAF crystal structure 1CI4 (helix α3, α4, α5 and α6) by minimizing the RMS deviation of the Cα coordinates of the extracted structures with respect to the ones of the reference structure. (**D**–**F**). Dispersion of the structures of the N-terminal region (residues 1 to 12) is represented by plotting the distribution histograms (in density mode) of the RMS deviations (RMSD) of the Cα of residues 1 to 12 with respect to the reference structure 1CI4. (**G**–**I**) Dispersion of the structures of the N-terminus (residues 1 to 4) is represented by plotting the same distribution histograms calculated on residues 1 to 4. (**J**–**L**) Dispersion of the structures of residues forming helix α1 (residues 5 to 12) is represented by plotting the same distribution histograms calculated on residues 5 to 12. The vertical red dotted lines show the median value of these distributions. (**M**) A representative frame extracted from the MD simulation of BAF Gly16Arg is displayed, with each BAF monomer being colored in light blue and green, respectively (N-terminal regions from residues 1 to 12 in dark blue and green, respectively). The Arg16 and Leu89 residues are displayed in ball-and-stick. The position of the MD simulation-identified salt-bridge is visualized by a black line between the guanidinium group of Arg16 of one monomer and the C-terminal carboxyl group of Leu89 of the other monomer. (**N**) Plots of the distance between the guanidinium group of Arg16 of one monomer (A, B) and the C-terminal carboxyl group of Leu89 of the other monomer (B, A) along one of the 1 µs MD trajectories. This distance is mostly constant along the three 1 µs MD trajectories calculated for BAF Gly16Arg.

**Figure 5 cells-12-00847-f005:**
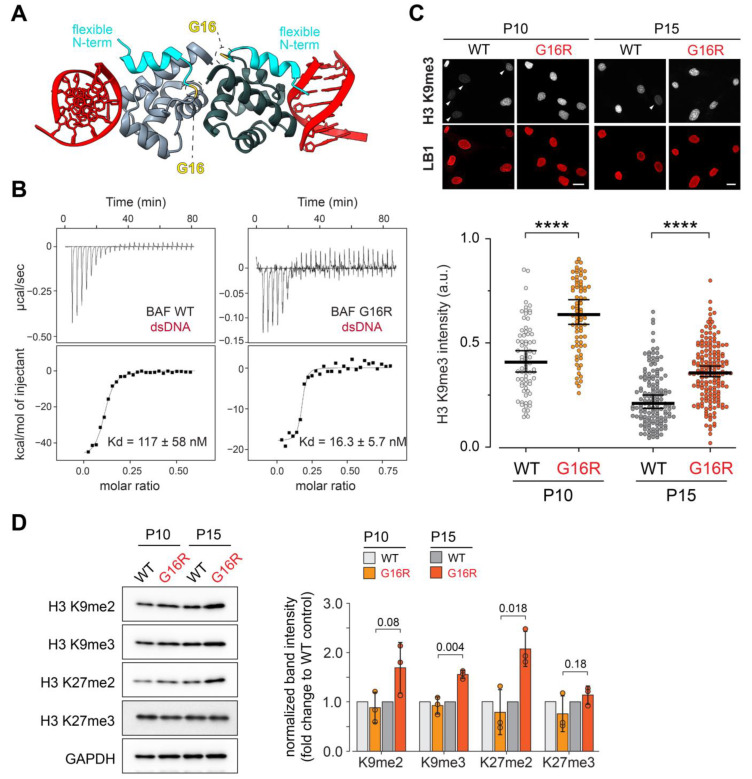
Enhanced DNA binding affinity of BAF Gly16Arg correlates with increased levels of repressed chromatin. (**A**) Cartoon of the crystal structure of the complex between the BAF dimer (light and dark grey) highlighting flexible N-terminal helix α1 (cyan) and dsDNA (red). The position of the mutated Gly in each BAF monomer is indicated. The PDB reference of this 3D structure is 2BZF. (**B**) ITC titration analyses providing the affinity of BAF WT and Gly16Arg for a 22 nt-dsDNA. The experiment was repeated twice, and the dissociation constants represent the mean values calculated from these experiments (see Appendix A). (**C**) **Top**: Confocal microscope images of control and patient passage 10 and 15 fibroblasts stained for histone H3K9me3 (white) and lamin B1 (LB1; red). Cells demonstrating low levels of H3K9me3 are indicated with arrowheads. Scale bars are 20 µm. **Bottom**: Graphs of changes in H3K9me3 expression intensity at P10 and P15. Measurements were made on a minimum of 66 nuclei per passage, (n = 2) using CellProfiler software [41]. Means were compared using unpaired *t*-test (Graphpad Prism), **** = *p* < 0.0001. (**D**) **Left**: Western blot analyses of proteins extracted from passage 10 and 15 control and patient fibroblasts. Blots were probed with antibodies against indicated histone modifications. GAPDH served as a loading control. **Right**: Quantification of the Western blot signal intensity was normalized to GAPDH. Values are shown as fold change relative to wild-type control; n = 3, error bars indicate S.D.; Exact values for paired two-tailed *t*-test are shown.

## Data Availability

The authors confirm that the data supporting the findings are available within the article and raw data are available from the first author upon reasonable request.

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
