# Peer review of "A De Novo Sequence Variant in Barrier-to-Autointegration Factor Is Associated with Dominant Motor Neuronopathy"

_cells, 2023, doi:10.3390/cells12060847_

Round 1

Reviewer 1 Report

Comments on the article cells-2229868 entitled: “A de novo sequence variant in Barrier-to-autointegration factor is associated with dominant motor neuropathy ” By Agathe Marcelot and collaborators.

The Authors identified in a patient with motor neuropathy a de novo and heterozygous mutation in BANF1 (c.46G>A [p.Gly16Arg]), encoding barrier-to-autointegration factor 1 (BAF). This novel mutation is located near the dimer interface and the interface used in binding the lamin A/C IgFold domain. Hence, this novel mutation is located four residues away from previously identified BAF Ala12Thr mutation that is associated to NGPS, a premature aging syndrome. To understand the consequences of BAF Gly15Arg mutant, the authors performed NMR and MD simulation. They show that BAF Gly16Arg substitution introduces a salt bridge between BAF monomers, that triggers the reduction in the flexibility and the conformation of the N-terminal region, thereby increasing the binding affinity to double-stranded DNA.

Specific comments

The structural NMR and MD analyses are convincing and supported by the results provided in this manuscript. 

Section: The Gly16Arg variant confers modest changes to nuclear lamina structure 

-Figure 2: Immunohistochemical localization of Lamin A/C, Lamin B1 and Emerin in BAF Gly16Arg mutant cells show normal distribution. However, the scoring of the number of dysmorphic nuclei shows an increase in BAF Gly16Arg mutant cells relative to normal cells. 

The levels of expression of BAF, lamin A/C, lamin B1 and emerin in BAF Gly16Arg mutant cells are missing. The authors should provide western blot analyses of these components. As indicated by the authors, loss of BAF in NGPS mutant cells and overexpression of BAF in transfected cells cause abnormalities to the nuclear envelope. Therefore, a western blot and an immunohistochemical localization of BAF in Gly16Arg mutant cells is critically needed to clarify this question.

Overall, this manuscript is very interesting and well presented. The majority of the conclusions are supported by the results presented in this manuscript.

Author Response

Reviewer #1

Thank you for your comments that our manuscript is very interesting and well presented.

Concern: A western blot and immunohistochemical localization of BAF in Gly16Arg mutant cells is critically needed to clarify this question.

Response: We agree that analysis of BAF would be of interest. However, we have not completed these experiments. We are concerned about the past history of western analysis with BAF. Extant antibodies are variant specific (Paquet et al., 2014; Lin and Engelman et al., 2003), which led to incorrect conclusions of the NGPS variant (Puente et al., 2011).  For this reason, a significant amount of work would be needed to validate extant antibodies that we do not even know whether they will recognize the G16R variant. As such, we have not completed these experiments for these studies.

Reviewer 2 Report

This is a well written paper which presents clear and and convincing data to support the authors hypothesis as to the effect of the G16R mutation on the properties of BAF. 

I have a couple of comments/questions

The microscope figures as presented were hard to interpret as there was no annotation to direct the reader to important regions. In particular I did not feel that figure 5c added anything that wasnt shown better by the western blot. 

I would also have liked to have seen a little more evidence in support of the authors suggestions of changes in heterochromatin eg What happens to non euchromatin histone markers do these decrease? What happens to the accessibility of the DNA as measured by techniques such as MNase digestion?  

Author Response

Reviewer #2

Thank you for your comments that our manuscript is well written and presents clear and convincing data.

Concern: The microscope figures, especially Figure 5C, were hard to interpret.

Response: We have added clarification to the figure legends. For Figure 2C, we noted that “An enlarged imaged of the boxed nucleus is shown on the right hand side of the panel.” For Figure 5C, we noted “Cells demonstrating low levels of H3K9me3 are indicated with arrowheads”.

Concern: What happens to non-euchromatin histone markers, do they decrease? What happens to accessibility of the DNA as measured by MNase digestion?

Response: We agree that these are interesting questions. However, we feel that these experiments are beyond the scope of these studies.